# Optimizing Road Safety Inspections on Rural Roads

**Giuseppe Cantisani** [1,*], **Chiara Caterina Borrelli** [2], **Giulia Del Serrone** [1] and **Paolo Peluso** [1]

1. Department of Civil, Constructional and Environmental Engineering, University of Rome La Sapienza, Via Eudossiana 18, 00184 Rome, Italy
2. Civil Engineer, 00100 Rome, Italy
* Correspondence: giuseppe.cantisani@uniroma1.it

**Abstract:** Road safety depends on a complex balance between all the factors that compose the road system: user, vehicle, environment, and infrastructure. Directive 2008/96/EC introduces procedures to improve the European Transport Network (TEN-T) safety, recognizing an important role for safety inspections, but focusing them on freeways and highways. This paper proposed methods and criteria useful to optimize current inspection procedures and adapt them also to secondary and local rural roads. It is often complex to determine the severity and weight of the different risk factors in this context, since deficiencies and faults can be found both in the road infrastructure and in the other components of the system. The proposed survey method was applied to a stretch of the SS7 Appia state road (Lazio, Italy). Producing video capture and using GPS sensors to geolocalize the data proved to be very useful for the compilation of the survey forms. The results showed a good consistency between the safety assessments obtained from the scoring of the inspection forms and the historical accident rate. Therefore, the proposed methodology can be a valuable aid in understanding safety concerns and in defining the improvement actions.

**Keywords:** rural roads management; safety inspections; survey forms; road safety improvement

## 1. Introduction

Road safety is a complex balance between the different components of the road system: user, vehicle, environment, and infrastructure. The resulting interaction can be termed "dynamic," since the user learns and processes information from outside at each instant and adjusts his or her driving behavior accordingly. Potential issues in the road system can disrupt the balance and reduce the safety margin that ensures the proper functioning of the system. Risk, in fact, can be caused by defects or deficiencies of the road infrastructure.

In Europe, the "road infrastructure safety management" has been codified in Directive 2008/96/EC [1], transposed in Italy into the Legislative Decree 35/2011 [2], and in subsequent amendments and additions (Legislative Decree 213/2021 [3]). Initially, the scope of the decree fell (Art.1, par.2) within the roads of the Trans-European Transport Network (TEN-T). To date, with the introduction of the Legislative Decree 213/2021 (transposition of the Directive 2019/1936/EC [4]), the standard is extended to other roads, with certain characteristics, and road infrastructure projects in rural areas. From 1 January 2025, the methodology will also be applied to the road network of national interest.

Recent regulations pursue the primary objective of assessing the safety of open roads to vehicular traffic through preventive analysis. Thus, investigations aim to preliminarily identify the technical, geometric, and functional characteristics of the road that may result in an accident [5] ("diagnosis phase") and subsequently identify the best strategies for mitigating or eliminating critical issues ("treatment phase"). This goal can be pursued in several ways.

Nowadays, the most implemented strategy involves in-situ inspections for an immediate reading of risk situations for traffic. Indeed, over the years, several research projects have tried to provide efficient guidelines for road inspections, among which is the

"Safety Prevention Manual for Secondary Roads" [6]. This manual, which is part of the "PILOT4SAFETY" pilot project financed by the European Commission, aims to train and support both "Road Safety Auditors" and "Road Safety Inspectors." Another example is the "Manual for Safety Inspections of Secondary and Local Rural Roads" [7] written within the IASP (Identification and Adaptation of Dangerous Roads) project, co-funded by the European Community and the Province of Catania (Italy).

Useful risk indices can be found in the literature to quantify user responsibility in road safety. Leur and Sayed (2002) developed a driver-based risk assessment criterion [8]. The process is based on a well-defined set of measurable road characteristics that are scored during the inspection. These scores are combined to obtain a risk index taking into consideration three essential factors: the user's exposure to road hazards, the user's likelihood of being involved in an accident, and the magnitude of the consequences in the event of an accident. In 2002, the Euro Road Assessment Program (EuroRAP) started, resulting into the Star Rating methodology in 2004, which processes data collected during a road inspection and quantifies the levels of safety offered by the road to the road users (drivers, motorcyclists, pedestrians, and cyclists), using a quantitative method. In this method, the road characteristics influence the severity or probability of occurrence of certain types of accidents (head-on collision, roadway departure) in which users may find themselves involved. The Star Rating Scores (SRS) risk index is then calculated by adding the individual sub-scores of these accident types according to cause-and-effect models [9,10]. Montella (2005) proposed a systematic procedure to determine which road characteristics are analyzed during the inspection and with which methods [11]: the procedure is based on historical accident rates, to obtain a Safety Improvement Index (PFI) which quantifies the level of safety achieved by solving the infrastructural deficiencies identified during the inspection. Cafiso et al. (2007) proposed a methodological approach for "two-lane rural highways" based on analytical procedures that analyze "design consistency" and a "Safety Index" (SI), which quantitatively measures the safety performance of the road segments inspection [12]. It is a combination of three components: accidental risk exposure, probability of occurrence, and consequences. The procedure is systematic and reproducible, regardless of the availability of accident data at the examined site. Another road safety performance index, the so-called Risk Index (RI), was developed by Vaiana et al. [13] to take into account the risk deriving from an infrastructure's features.

Through safety inspections, it can be shown how certain infrastructure features may affect the occurrence of accidents. To assess, for example, the effects of the absence or the inadequacy of pavement width on accident occurrence on single-carriageway suburban roads, results obtained in a study by Zegeer and Council (1995) showed that adequate lane and shoulder widths significantly improved accident rates [14]. Harwood et al. (2000) [15], on the other hand, showed that roads where paved shoulders are absent and traffic flows are between 500–2500 vehicles/day tend to have 10 to 50 percent more accidents when compared to roads with a 1.8 m paved shoulder.

More generally, it has been observed that certain types of accidents, combined with infrastructure deficiencies, may suggest the adoption of specific countermeasures. Some studies [16–18] investigating the causes of accidents, have identified some recurring factors as possible explanations leading to certain accident types. With reference to front-to-side and/or head-on collisions [19], which are generally the most serious accidents and often result in serious or fatal consequences for vehicle occupants, these studies identified some potentially critical factors for rural roads with a single carriageway and one lane for each travel direction. Among these were an insufficient passing sight distance, repeated small radius curves, inadequate lane widths and road accesses, poor road markings, and the absence of shoulders.

For rear-end crashes [20], on the other hand, possible infrastructure-related causes include: levels of service characterized by interrupted flow conditions and frequent "stop-and-go" events (e.g., sudden stops on roads during busy periods), poor pavement adherence, improperly signalized intersections, or the presence of roads converging at intersections

characterized by steep slopes, the presence of commercial activities, and parking areas along the road. Finally, roadway departures [21] can occur with narrow lane widths, poor pavement adherence, tight curves not properly judged by the user, lack of roadway consistency, insufficient longitudinal superelevation, and failure to use the shoulder as a space for correcting the driver's trajectory.

In 2012, in implementation of the Legislative Decree 35/2011, the Guidelines for Road Infrastructure Safety Management were issued [22]. They established the criteria and methods to be adopted for carrying out the activities governed by the Decree 35/2011, collectively referred to as "road safety analyses".

Another strategy to perform safety-focused analyses, which has been developing in recent years, involves the investigation of the actual traffic operating conditions of a road network by analyzing big data, such as Historical Car Data (HCD) [23,24]. HCD are a useful tool for reliably reading sample trends of operational speeds along a road section, and for identifying the factors influencing the driving behavior of different road users [25].

Since most of the criteria and procedures proposed in the literature, and developed in technical practice, mainly concern freeways and arterial roads, there is a certain lack of knowledge and methodologies for collectors and local roads. However, there are some ongoing research studies that should be mentioned, which have undertaken safety investigations on rural roads [26].

In this paper, the authors proposed useful methods and criteria for optimizing road safety inspections on secondary and local rural roads. The practical use of the methodology was presented through a case study related to the state highway S.S.7- Strada statale Appia. In detail, the proposed method involves inspection activities to be carried out through semi-automated procedures with low-cost instrumentation and technology easily accessible by all inspectors or analysts. The risk and severity assessment related to deficiencies in infrastructure elements was carried out by assigning weights in the inspection reports. In addition, a single quantifying index was introduced to summarize the outcome of the analysis. Finally, the comparison between the outcomes of the reports and the historical accidents recorded at the survey site assessed the sensitivity of the method with respect to the possibility of predicting potentially hazardous conditions. The results could be used to plan a series of repair and maintenance interventions to be suggested to the managing authority to ensure high safety standards of infrastructures.

## 2. Materials and Methods

This paper proposed an innovative methodology to optimize traditional road inspection activities by obtaining a single value that quantifies the degree of safety and risk exposure of an infrastructure. It does not claim to substitute traditional safety inspections, but it involves automation in the inspection processes, using most advanced instrumentation to simplify and streamline the conducted work. In fact, in addition to the current inspection techniques, the authors associated the survey reports with original and worthwhile tools to be presented and discussed. The innovative and constituent elements of the proposed method are as follows:

- The inspection reports, arranged just for freeways and major highway roads, are applied to secondary and local rural roads. These types of roads present different safety issues related to the arterial ones. In fact, they are characterized by greater promiscuity both in terms of the type of road users and the interaction with the territory. This is due to their greater accident unpredictability, caused by less defined mobility functions;
- The integration of theoretical evaluation criteria of the road (e.g., geometrical, and technical) in the settlement of an investigation inspection process;
- Video recording while driving and the subsequent processing through the application of calculation codes improves the readability of the infrastructure because inspectors can consult videos synchronized with the location of the vehicle in relation to the road distances;

- Weights and scores were proposed for the different characteristics of the infrastructure, which are sensitive to the type of the performed analysis and the roadway inspected.

In this way, the outcome of the inspections is not based on the experience and sensitivity of the inspector, due to his/her judgment uniformity.

Road safety inspections can be divided into two macro-phases: the diagnosis phase and the treatment phase. The following is a description of the objectives of the two phases and a proposal of the methods and criteria that can optimize inspection procedures when performed on secondary and local rural roads.

### 2.1. Diagnosis Phase

The diagnosis phase is useful for assessing the safety of the under-study infrastructure through the completion of the inspection reports adapted from those available online [27]. It is composed of two consequential steps: the preliminary analysis and the general inspection, as provided in the Guidelines taken as reference [22]. The preliminary analysis is the time when all preparatory and necessary activities for the completion of the general inspection are arranged.

### 2.1.1. Procedure for the Identification of the Existing Road Alignment

The design drawings and the document archive of a road network are useful tools for understanding the safety issues related to the layout of secondary and local rural roads [28]. To overcome the challenge of cadastral information access of an existing road network, a method already proposed in the literature can be adopted. Using a computational code, the horizontal alignment of a road axis can be reconstructed from a georeferenced graph [29], with a consequent processing time reduction.

### 2.1.2. Use of GPS Sensors and Videos

To simplify the on-site report-filling phase and to protect the user's safety, it is effective to use videos [30] and GPS sensors as visual and informational support for the survey.

Capturing videos through a dashcam integrated into the vehicle is useful to record GPS coordinates and the vehicle's speed at any instant, in addition to the multimedia data. Moreover, the use of a smartphone application allows for the acquisition of position (latitude, longitude, altitude, speed, and heading), acceleration, angular velocity, and three-axis orientation data. Through this data, videos can be generated to visualize the location of the vehicle along the roadway in a general framing, in a detailed framing (useful for identifying intersections), and in an elevation profile (to define the correspondence between the issues found on videos and the vertical alignment). Through one last video, it is possible to punctually show the lines of sight, based on the theoretical speed diagram required to stop the vehicle safely. These videos are synchronized to the video captured by the dashcam by matching the GPS coordinates. In this way, it is possible to obtain a unique video that provides, at each second, a comprehensive and objective sight of the infrastructure issues.

### 2.1.3. Reports Compilation

The safety assessment of the investigated roads starts by compiling the safety reports, which are divided into general inspections—one for each direction of travel—and spot checks. The state highway S.S.7 under study was divided into sections of 500 m in length. Both general and punctual scores were given to each section, regarding various technical and functional aspects of the infrastructure, and were classified into Macro-items, Items, Parameters, and Indicators.

The main macro-items analyzed in the general inspections are as follows: general aspects, roadway, road marking, access, pavement, and other aspects, an excerpt of which is given in Table 1.

**Table 1.** Reports distribution in Macro-items, items, parameters, and indicators for general inspections.

| Macro-Items | Items | Parameters | Indicators |
|---|---|---|---|
| Right-of-way | (Roadway, Median, and Roadside) | Shoulder | Suitable width or absence |
| | | | Shrinkage near a structure |
| | | Lane and Fast Lane | Suitable width |
| | | | Excess width |
| | | Sideslope | Building maintenance |
| | | | Green maintenance |
| | | Drainage | Maintenance |
| | | Fencing | Maintenance |

For spot checks, on the other hand, there is a single macro-item called "At-grade intersections/Accesses". The indicators refer to any intersection falling within each analyzed section of 500 m length.

An integer between 1 and 3 is assigned to each indicator, subsequently referred to as $a_i$ for each $i$-th indicator, in proportion to the degree of risk of the analyzed issue. The degree of judgment is assigned based on qualitative assessments of risk, multiplying, for example, the severity of the problem encountered and its frequency. A value equal to 0 is used in cases where an aspect in the reports is not detectable due to location inaccessibility, impossibility of execution, or lack of data and documents provided by the managing bodies or not available from reliable sources (e.g., quantity and type of traffic). With respect to certain road types, some indicators cannot be considered, and the abbreviation "N" is used.

In the present case study, some macro-items, items, parameters, and indicators do not match the road type analyzed. In fact, they refer to aspects that can only be found on freeways and highways (e.g., emergency lane suitability, inner shoulder, median, interchanges characteristics). Since it is not possible to fill the forms with such information, these kinds of indicators have been removed from the reports, resulting in no change to the outcome. In addition, other indicators were modified according to a different assessment method, e.g., instead of the IRI and CAT index values to evaluate the pavement condition, a visual and qualitative analysis was performed to identify deformations, drainage capacity, and adherence performance.

*2.2. Treatment Phase*

The treatment phase includes the elaboration of the analysis report, which involves a summary of the diagnosis phase and the subsequent identification of interventions to mitigate and eliminate the identified risks.

2.2.1. Report Indicators Weightings Proposal

Reference can be made to the "Safety Cube" project [31] for the indicator weightings in the reports. In particular, "Deliverable 5.1, Identification of infrastructure-related risk factors" [32] collects, for each indicator named in the "Safety Cube" project risk factor, a synopsis that outlines their main results through a process of meta-analysis. The project provides assigns each risk factor a color code, as shown in Figure 1:

"The colour code indicates how important this risk factor is in terms of the amount of evidence demonstrating its impact on road safety as regards increasing crash risk, frequency or severity. The following codes and definition were applied:

- Red: Risky. Consistent results showing an increased risk of crashes or injuries when exposed to this risk factor;
- Yellow: Probably risky. Some evidence that there is increased risk when exposed to this risk factor, but results are not consistent. This could be because while the majority

of studies demonstrate a risk, there may be some studies with inconsistent results. Or, studies indicate a risk but are few in number or have methodological weakness;

- Grey: Unclear. Studies report opposite effects. There are few studies with inconsistent results, few studies with weak indication or risk;
- Green. Probably not risky. Studies consistently demonstrate that this risk factor is not associated with increased crash risk, frequency or severity."

| Red (Risky) | Yellow (Probably risky) | Grey (Unclear) |
|---|---|---|
| ! Effect of Traffic Volume on safety | ! Occurrence of Secondary crashes | ? Congestion as a risk factor |
| ! Risks associated with Traffic Composition | ! Alignment deficiencies - Absence of Transition curves | ? Risks associated with the distribution of traffic flow over arms at junctions |
| ! Road Surface - Inadequate Friction | ! Risk of Different Road Types | ? Adverse weather - Frost and snow |
| ! Workzone length | ! Adverse weather - Rain | ? Workzone duration |
| ! Alignment deficiencies - Low Curve Radius | ! Poor Visibility - Darkness | ? Alignment deficiencies - Frequent curves |
| ! Cross-section deficiencies - Number of Lanes | ! Cross-section deficiencies - Superelevation | ? Alignment deficiencies - Densely spaced junctions |
| ! Shoulder and roadside deficiencies -Absence of paved shoulders | ! Alignment deficiencies - High grade | ? Interchange deficiencies - Acceleration / deceleration lane length |
| ! Shoulder and roadside deficiencies - Narrow Shoulders | ! Presence of Tunnels Cross-section deficiencies - Narrow lanes | |
| | ! Undivided road | |
| | ! Cross-section deficiencies - Narrow median | |
| | ! Shoulder and roadside deficiencies - Risks associated with Safety Barriers and Obstacles | |
| | ! Shoulder and roadside deficiencies - Sight Obstructions (Landscape, Obstacles and Vegetation) | |
| | ! Interchange deficiencies - Ramp Length | |
| | ! At-grade junctions deficiencies - Number of conflict points | |
| | ! Risk of different junction types At-grade junction deficiencies - Skewness / Junction angle | |
| | ! At-grade junction deficiencies - Poor sight distance | |
| | ! At-grade junction deficiencies - Gradient | |
| | ! Uncontrolled rail-road crossing | |
| | ! Poor junction readability - Absence of road markings and crosswalks | |
| | ! Poor junction readability - Uncontrolled junction | |

**Figure 1.** Infrastructure related risk factors synopses by colored code [32].

In the present case, the indicator weightings, denoted by $p_i$ for each $i$-th indicator, were carried out by the following method: parameters or items with risk factors classified as red are weighted 1, risk factors classified as yellow are weighted with a value of 0.67, in all other cases the weight assigned is 0.33.

### 2.2.2. Report Synthetic Value Proposal

In order to define the weights assigned to the different infrastructure risk factors in the reports, a single synthetic value was calculated for each analyzed section, that was able to summarize the inspection outcomes. In this way, it is possible to search for a correspondence between the values obtained and the historical accident data, to be able to understand whether the sections characterized by serious infrastructure deficiencies are more exposed to the accident phenomenon.

Total scores are calculated for each $j$-th section of the analyzed road by summing the scores of the general reports, for the two travel directions, and the spot checks. The total scores associated with the general inspections ($TS_{GI,j}$) are defined through the product between the ratings $a_{ij} > 1$ and the corresponding weights $p_i$, with n number of indicators:

$$TS_{GI,j} = \sum_{i=1}^{n} a_{ij} * p_i \tag{1}$$

For the punctual inspections, spot checks ($\overline{TS_{PI,j}}$), the product between the $a_{ij}$ ratings and the $p_i$ weights is defined first, and then, if there are multiple intersections and/or accesses within the same section, a single score is defined by averaging the scores of the $m$ intersections falling within it:

$$\overline{TS_{PI,j}} = \frac{\sum_m \sum_{i=1}^{n} a_{ij} * p_i}{m} \tag{2}$$

The final single synthetic value called "Normalized Total Score" ($TS_{norm\_j}$) is calculated for each $j$-th 500 m length section by summing the scores of the two general reports (in the two travel directions) and the score of the spot checks. This value is, then, normalized in relation to the maximum total score $TS_{max}$ obtained along the entire analyzed road arc:

$$TS_{norm\_j} = \frac{TS_{GI1,j} + TS_{GI2,j} + \overline{TS_{PI,j}}}{TS_{max}} \tag{3}$$

In order to assess a respondence between the infrastructural deficiencies and the historical accident data, the $TS_{norm\_j}$ was compared with the recorded incident provided by Istat (Italian National Institute of Statistics) and Aci (Automobile Club of Italy) [33], normalized in relation to the maximum detected value ($ACC_{norm\_j}$). Finally, the introduction of some thresholds allowed for the quantities categorization and their representation through a chromatic scale, presented in Figure 2:

| $TS_{norm\_j}$ | $ACC_{norm\_j}$ | Normalized parameter |
|---|---|---|
| Low | Low | < 0.40 |
| Medium | Medium | 0.40 ÷ 0.60 |
| High | High | 0.60 ÷ 1.00 |

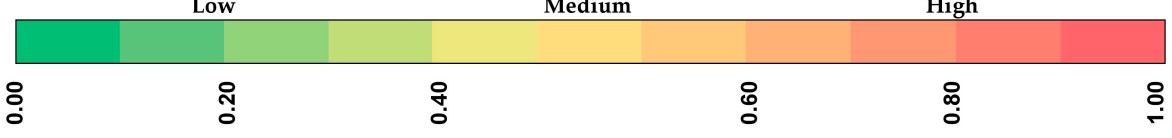

**Figure 2.** Normalized Total Score and Normalized Accident Rate Chromatic scale.

The comparison between the results of the reports and the historical accident data is useful to propose, for road sections characterized by a high level of respondence, countermeasures to mitigate or eliminate the risks found. Moreover, these proposals could be more effective identifying the most frequent accident modes and the categories of users most affected.

### 2.3. Case Study

The proposed methodology was tested on the state highway S.S.7- Strada statale Appia. The section investigated is between the distance 30 + 450 km (Genzano di Roma) and 38 + 045 km (Velletri) and it is managed by ANAS S.p.A., an Italian company deputed to the construction and maintenance of Italian motorways and state highways. Two inspections were carried out: a daytime inspection to better examine the road section and to acquire images at specific problematic points; and an evening inspection to assess the conditions of artificial lighting and the presence of problems that, in the presence of natural light, would not have emerged. The reports compilation was done a posteriori by examining the post-processed videos (Figure 3) and assigning judgments to the macro-items as suggested by the specifications and technical manuals cited [6,7].

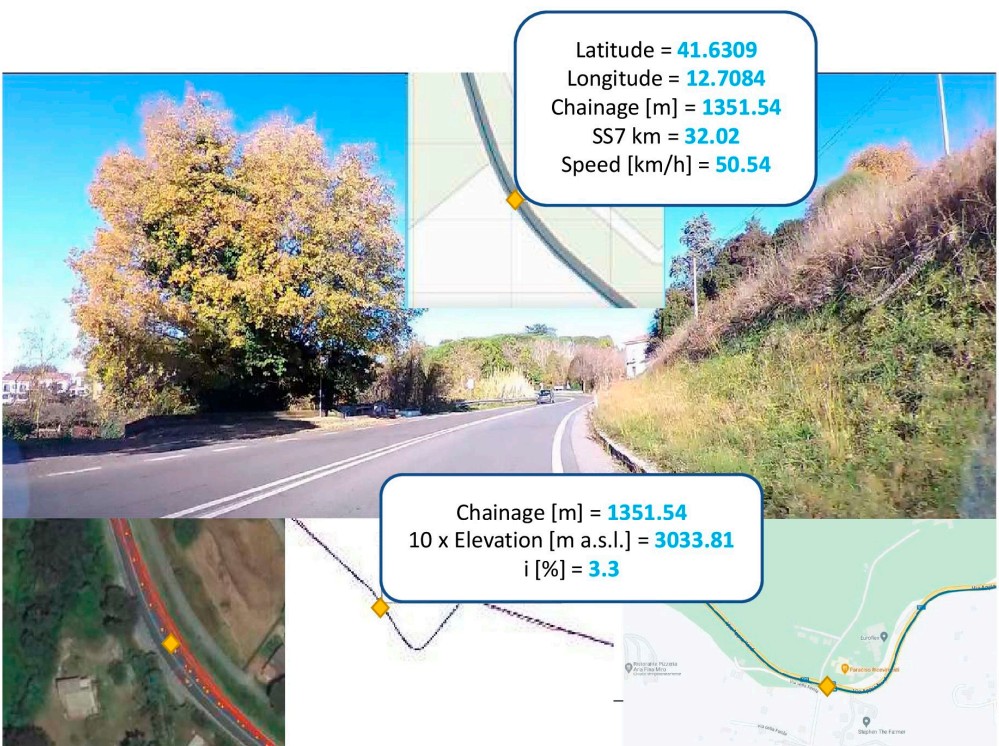

**Figure 3.** Videos synchronization to conduct the inspections.

### 3. Results

The identification of the geometric elements of the reference axis of the existing road, shown in Figure 4, permitted the application of the regulatory model for the design verification, required by the Italian Ministerial Decree dated 5 November 2001 [34], shown in Tables 2 and 3 (horizontal and vertical alignment verifications, respectively).

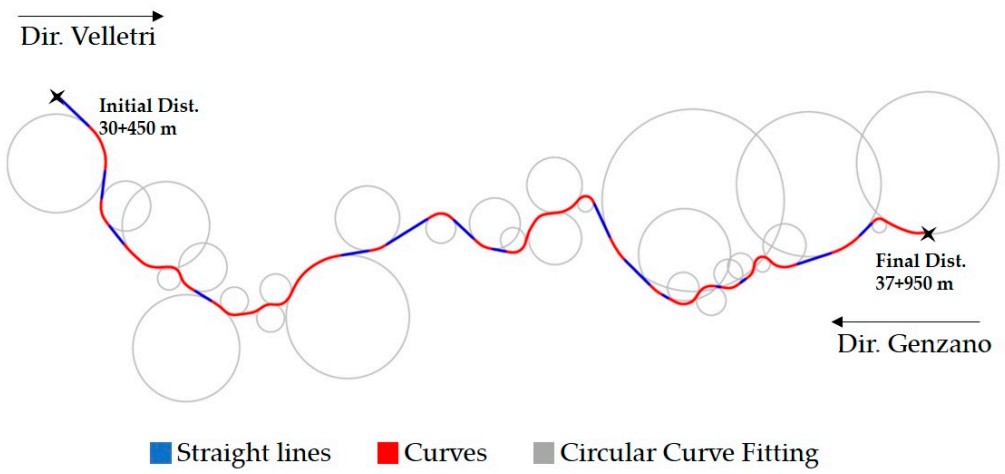

**Figure 4.** Identification of the existing road alignment.

**Table 2.** Geometric verifications related to the road under-study-Horizontal alignment.

| | | Straight Road | | Circular Curves | | | |
|---|---|---|---|---|---|---|---|
| Initial Dist. (km) | Final Dist. (km) | $L > L_{min}$ | $R > R_{min}$ | $\Delta V = V_i - V_{i+1}$ Velletri Dir. (km/h) | $\Delta V = V_{i+1} - V_i$ Genzano Dir. (km/h) | $D_T \leq D_R$ Velletri Dir. | $D_T \leq D_R$ Genzano Dir. |
| 30.45 | 30.95 | X | | | | | |
| 30.95 | 31.45 | | | | | | |
| 31.45 | 31.95 | | | $V_3 - V_4 = 33 > 20$ | $V_5 - V_4 = 24 > 20$ | | |
| 31.95 | 32.45 | X | | $V_6 - V_7 = 37 > 20$ | $V_{10} - V_9 = 40 > 20$ | | |
| 32.45 | 32.95 | | | | | | |
| 32.95 | 33.45 | | | $V_{10} - V_{11} = 22 > 20$ | | | |
| 33.45 | 33.95 | | | | | | |
| 33.95 | 34.45 | X | X | | | | |
| 34.45 | 34.95 | | | | | | |
| 34.95 | 35.45 | | | $V_{19} - V_{20} = 28 > 20$ | $V_{18} - V_{17} = 58 > 20$ | | 395.68 > 333 |
| 35.45 | 35.95 | | | | | | |
| 35.95 | 36.45 | X | | | | | |
| 36.45 | 36.95 | X | X | | $V_{25} - V_{24} = 22 > 20$ | | |
| 36.95 | 37.45 | | | | $V_{26} - V_{25} = 36 > 20$ | | |
| 37.45 | 37.95 | X | X | $V_{26} - V_{27} = 60 > 20$ | $V_{28} - V_{27} = 60 > 20$ | 368 > 321 | 403 > 333 |

**Table 3.** Geometric verifications related to the road under-study-Vertical alignment.

| | | Vertical Curve Fitting | | Visibility | | Plano-Altimetric Coordination |
|---|---|---|---|---|---|---|
| Initial Dist. (km) | Final Dist. (km) | $R_v > R_{vmin}$ | $\Delta_{max}$ | $\Delta_{max}$ | Obstacles | |
| 30.45 | 30.95 | | −6.1 | −1.6 | Sideslope Curve 1 | |
| 30.95 | 31.45 | | | | | |
| 31.45 | 31.95 | | 3.8 | 1.3 | Sideslope Curve 3 | |
| | | | −2.8 | 0 | Vegetation Curve 4 | |
| 31.95 | 32.45 | | 0 | 3.2 | Sideslope Curve 5 | |

**Table 3.** *Cont.*

| | | Vertical Curve Fitting | | Visibility | | | Plano-Altimetric Coordination |
|---|---|---|---|---|---|---|---|
| Initial Dist. (km) | Final Dist. (km) | $R_v > R_{vmin}$ | $\Delta_{max}$ | $\Delta_{max}$ | Obstacles | | |
| 32.45 | 32.95 | | 0 | 3.1 | Vegetation. Curve 9 | | |
| | | | −4.3 | −1.7 | Vegetation. Curve 10 | | |
| 32.95 | 33.45 | | 0 | 3.3 | Vegetation Curve 11 | | |
| 33.45 | 33.95 | | −3.4 | 0 | Barrier Curve 12 | | |
| 33.95 | 34.45 | | 0.8 | 4.4 | Wall Curve 13 | | |
| 34.45 | 34.95 | | −7.9 | −3.3 | Sideslope with Vegetation Curve 15 | | |
| 34.95 | 35.45 | | −2.4 | 0 | Sideslope with Vegetation Curve 17 | | |
| 35.45 | 35.95 | | 0 | 4.7 | Wall Curve 18 | | |
| | | | 0.5 | 1.5 | Sideslope Curve 19 | | |
| 35.95 | 36.45 | | 1.4 | 5.6 | Wall Curve 20 | | |
| | | | −3.5 | 0 | Trees Curve 21 | | |
| 36.45 | 36.95 | | 0.8 | 4.1 | Vegetation Curve 22 | | |
| | | | 0 | 2.2 | Wall Curve 23 | | |
| 36.95 | 37.45 | Sag:2000 < 2182 | 0 | 4 | Vegetation Curve 25 | | dir. Genzano 37.42: False fold of the perspective view of the roadsides |
| | | | 0 | 1.1 | Wall Curve 26 | | |
| 37.45 | 37.95 | Crest:1000 < 3188 | −2 | 0 | Barrier Curva 27 | | dir. Genzano 37.47: Masking of planimetric direction change |
| | | | 0 | 9.1 | Sideslope/wall Curve 28 | | |

where:

- $L_{min}$ is the minimum length for a straight line, depending on the maximum design speed. In the specific case $L_{min}$ = 150 m;
- $R_{min}$ is the minimum radius value for a specified road class. It depends on the minimum design speed of that class, and in the specific case $R_{min}$ = 118 m;
- $\Delta V$ is the speed difference between two adjacent elements, characterized by their own design speed;
- $D_T$ (transition distance) is the length in which the speed, according to the accepted theoretical model, passes from the value $v_i$ to $v_{i+1}$, of two consecutive elements;
- $D_R$ (recognition distance) is the maximum length of a road section within which the driver can recognize possible obstacles;
- $Rv_{min}$ is the minimum radius value for a vertical curve, related to the design criteria (i.e., geometric, dynamic, and sight distances verifications);
- $\Delta_{max}$ curve widening due to sight distances verifications.

In particular, the sight distance checking was carried out by means of an original procedure, subsequently implemented in a programming platform, which allowed for a rapid estimation of the possible pavement widenings, inside the curves, necessary to ensure safe conditions for driving (Figure 5).

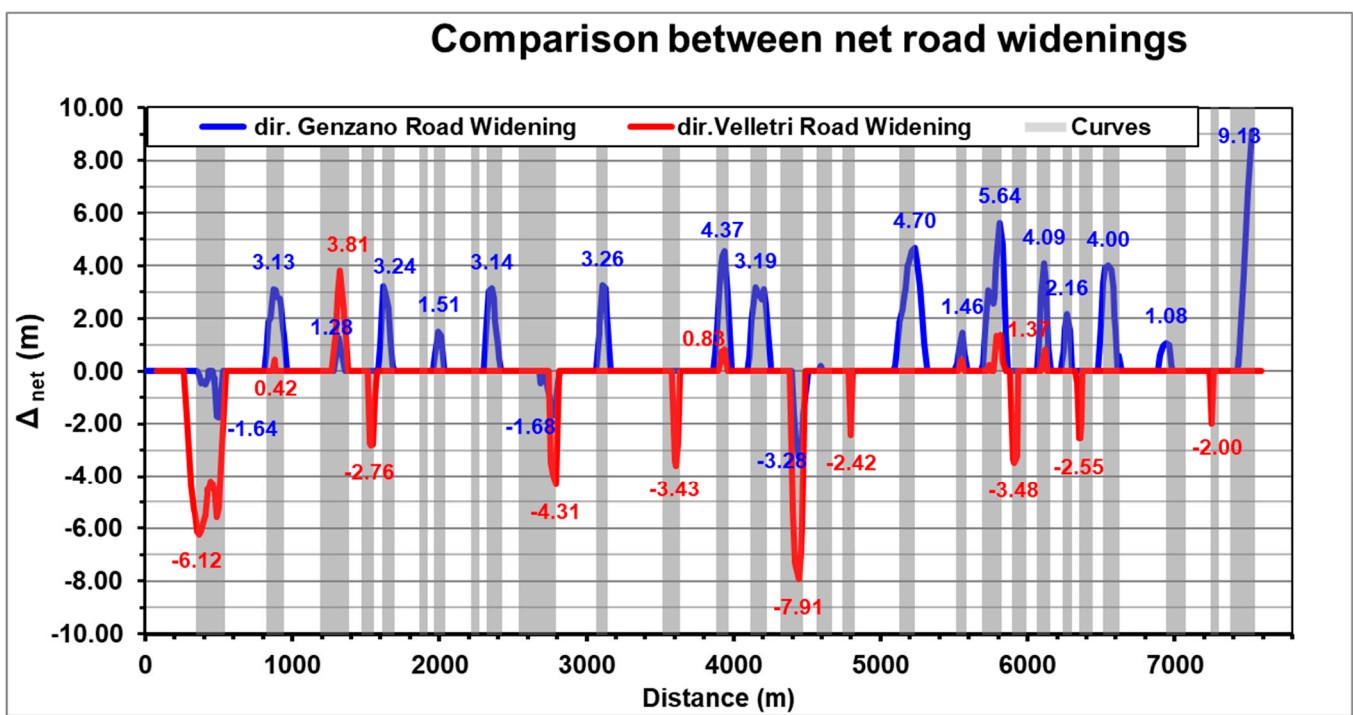

**Figure 5.** Road widenings in the two travel directions of the S.S.7.

The outcome of the inspection evaluations, summarized through the synthetic value $TS_{norm\_j}$, shows the sections where infrastructural deficiencies and issues result in risky conditions in terms of road safety. Risk is therefore associated with:

- the severity of the issue $a_{ij}$ (through the degrees of risk attributed to the indicators);
- the potential to result in a higher accident frequency $p_i$ (by means of the weights associated with the risk factors).

The accident rate was assessed on data published by Istat for the five-year period 2015–2020, which reported 85 accidents in the surveyed stretch of road, of which nearly 24% (20 accidents) were concentrated between the distance 32 + 000 km and 33 + 000 km, 20% (15 accidents) in the last km of the section, and 15% (14 accidents) in the second last km.

A summary of the outcomes of the reports and the accident rates is shown in Table 4, where the colored rows follow the colors of Figure 2:

A comparison of historical accident data and inspections identified 4, 13, 14 and 15, red colored in Table 4, as the most critical sections. On these sections, the infrastructure deficiencies have probably affected traffic safety, thus favoring the highest recorded accident density, and, therefore, the analysis of possible countermeasures to be investigated focuses on them.

Figure 6 shows a significant correspondence between the outcomes of the inspection assessments and the historical accident rate:

Figure 7, for example, shows 10 accidents per frontal-lateral and/or lateral collision, with reference to the road sections 4 and 13.

**Table 4.** A summary of the outcomes of the inspection reports.

| 500 m Section | 1 | 2 | 3 | 4 | 5 | 6 | 7 | 8 | 9 | 10 | 11 | 12 | 13 | 14 | 15 |
|---|---|---|---|---|---|---|---|---|---|---|---|---|---|---|---|
| Dist. Initial (m) | 30 + 450 | 30 + 950 | 31 + 450 | 31 + 950 | 32 + 450 | 32 + 950 | 33 + 450 | 33 + 950 | 34 + 450 | 34 + 950 | 35 + 450 | 35 + 950 | 36 + 450 | 36 + 950 | 37 + 450 |
| Dist. Final (m) | 30 + 950 | 31 + 450 | 31 + 950 | 32 + 450 | 32 + 950 | 33 + 450 | 33 + 950 | 34 + 450 | 34 + 950 | 35 + 450 | 35 + 950 | 36 + 450 | 36 + 950 | 37 + 450 | 37 + 950 |
| $TS_{GI,j,Velletri}$ | 15.02 | 13.01 | 18.99 | 23.01 | 15.03 | 16.02 | 16.00 | 17.68 | 20.64 | 19.03 | 23.69 | 23.34 | 20.68 | 19.03 | 22.03 |
| $TS_{GI,j,\_Genzano}$ | 16.02 | 11.33 | 18.31 | 18.69 | 8.99 | 12.33 | 13.32 | 16.67 | 15.00 | 19.68 | 22.34 | 24.69 | 22.69 | 21.02 | 25.37 |
| $TS_{PI,j}$ | 0.00 | 0.00 | 10.72 | 20.77 | 0.00 | 0.00 | 0.00 | 17.42 | 0.00 | 14.74 | 0.00 | 0.00 | 19.77 | 26.80 | 21.44 |
| $TS_{norm\_j}$ | 0.45 | 0.35 | 0.70 | **0.91** | 0.35 | 0.41 | 0.43 | 0.75 | 0.52 | 0.78 | 0.67 | 0.70 | **0.92** | **0.97** | **1.00** |
| ACC/km | 4.00 | 7.00 | | 20.00 | | 5.00 | | 11.00 | | 9.00 | | 14.00 | | 15.00 | |
| $ACC_{norm\_j}$ | 0.20 | 0.35 | | **1.00** | | 0.25 | | 0.55 | | 0.45 | | **0.70** | | **0.75** | |

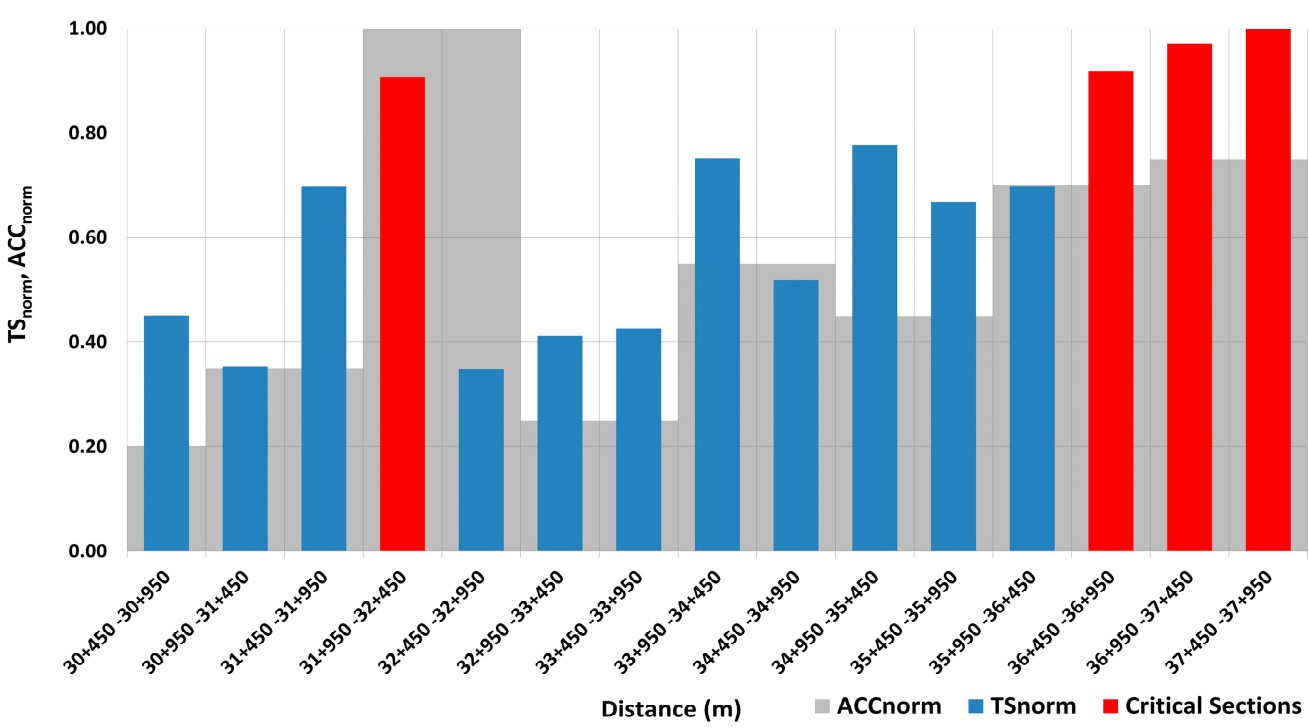

**Figure 6.** Comparison between the outcomes of the inspection assessments and the historical accident rate.

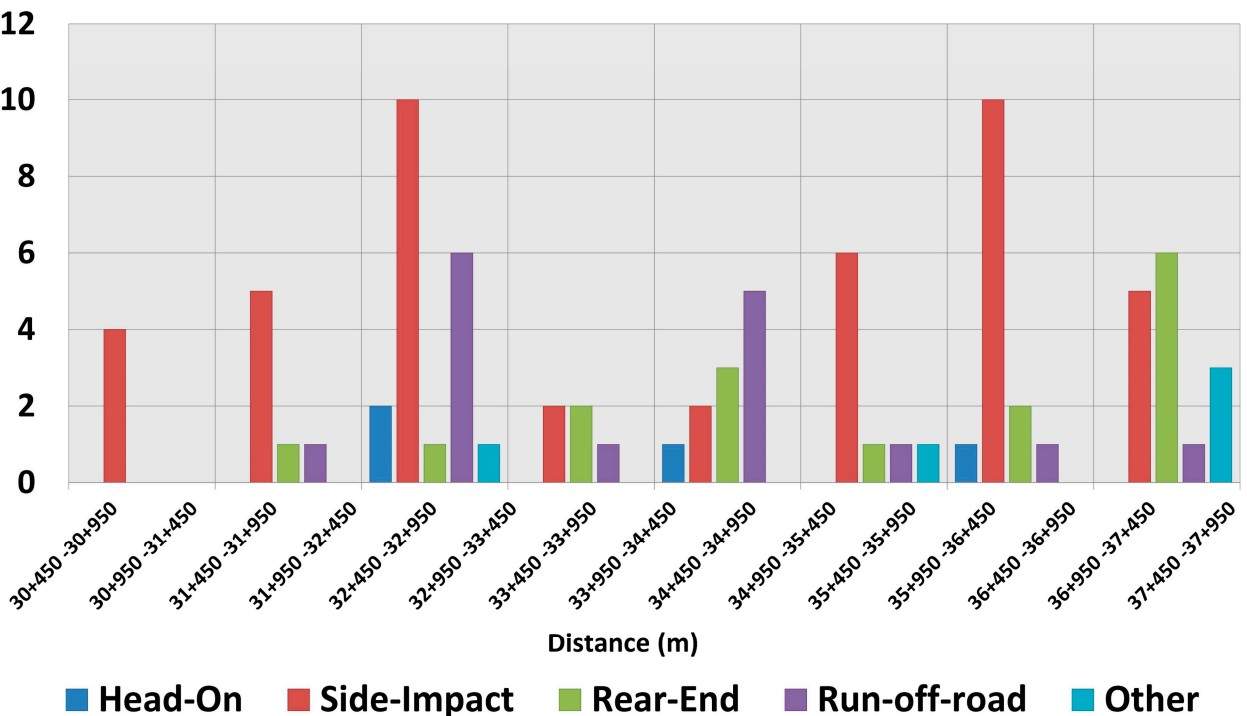

**Figure 7.** Accident types in the road sections.

## 4. Discussion

Appendix A contains summary tables of the general and punctual inspections carried out on each section of the examined road, with the scores related to each indicator and

their weights; through this tool, it is therefore possible to recognize the types and extent of infrastructure deficiencies found.

With regard to road Section 4, for example, in Velletri direction, the width of the shoulders is inadequate, there is a high density of direct accesses localized even on curves, and a lack of visibility on curves. The pavement, in both directions, is characterized by bleeding and smooth surface texture, factors that can promote a low supply of adherence. During the inspections, the presence of deteriorated and inefficient safety barriers at 32 + 230 km, frequent breaks in safety devices, inadequate terminals, and the absence of safety barriers on medium height embankments at 32 + 430 km were noted. With regard to punctual inspections, on the other hand, several accesses on curves (Via Poggio d'oro and Via Panoramica) were found, often characterized by a lack of visibility at the intersection/access due to obstructions located within the driveway for both priority and non-priority maneuvers. The absence of lighting and poor visibility in nighttime conditions were also noted, worsened by inadequate road markings. Finally, limited visibility was recorded in the entry onto Via Appia from Via Appia Vecchia (S.P. 85) due to an incident angle between the axes of the roads of less than 70°.

Furthermore, sections 14 and 15 are characterized by rear-end collisions as the most recorded accident type; this kind of accident could be determined by a sequence of high radii of curves to significantly lower radii, in Velletri direction. In Table 2, in fact, between 37 + 450 km and 37 + 950 km there is a difference between theoretical speeds ($\Delta V$ 28–27) equal to 60 km/h, with poor visibility conditions and the presence of an access within the curve, not adequately marked.

Some of the critical issues found along the road can be observed in Figure 8:

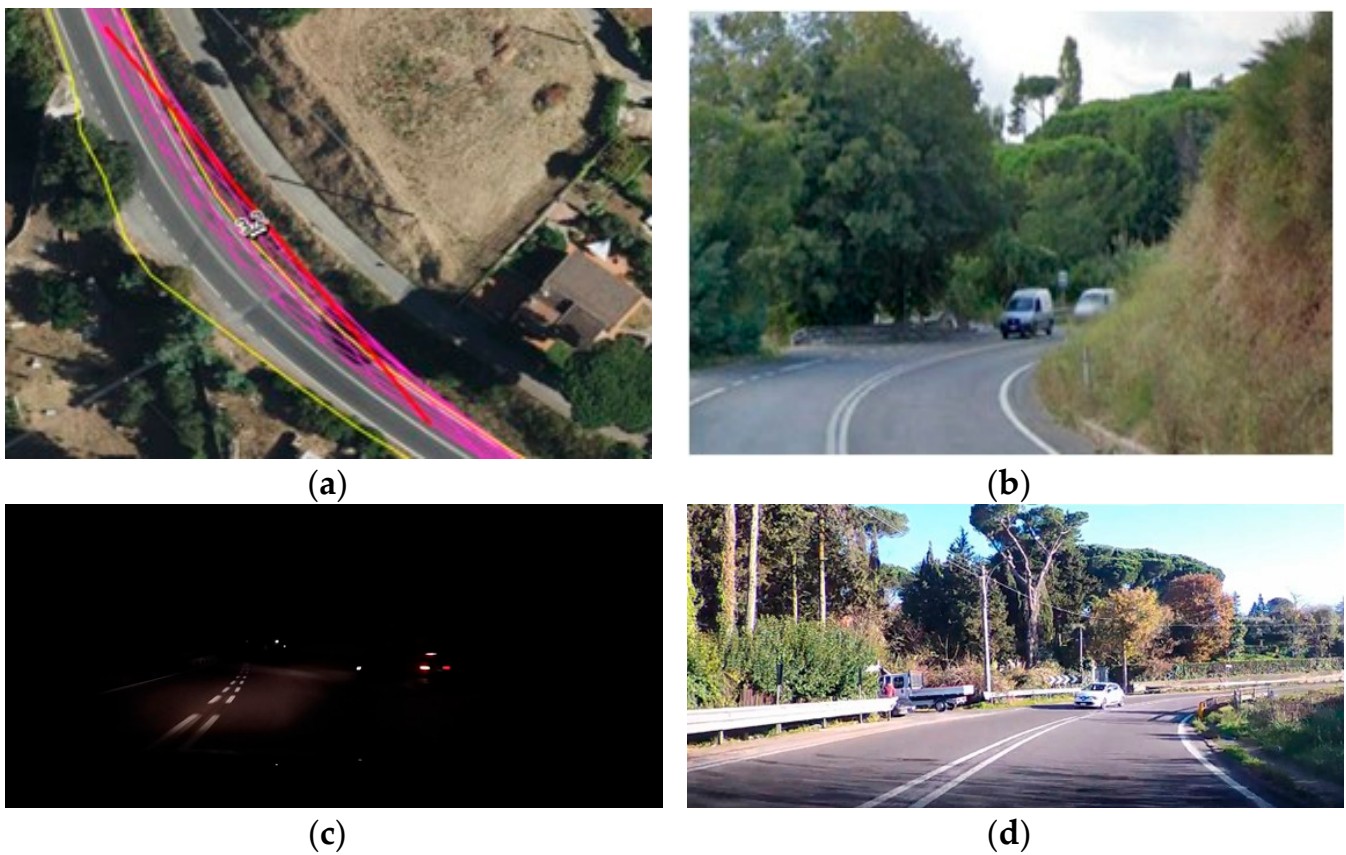

**(a)**          **(b)**

**(c)**          **(d)**

**Figure 8.** Examples of issues found along the road: (**a**,**b**) poor visibility in curves, (**c**) problems with road markings visibility at night times, and (**d**) problems related to frequent interruptions of safety devices, inadequate terminals, and repeated accesses on curves.

In accordance with the scientific literature, due to the problems found, some useful countermeasures can, therefore, be proposed to eliminate or mitigate the negative effect of the infrastructure deficiencies. Several studies have recommended more cautious behavior of road users in the case of roads with narrow lanes and shoulders, as the driver is induced to reduce speed [35]. On the other hand, however, narrow shoulders may cause drivers to deviate their trajectory to the inside of the roadway, risking a head-on collision with oncoming vehicles, especially for rural roads where a median strip is absent [36]. Moreover, a shoulder width greater than 0.5 m allows users to perceive obstacles farther away and feel safer assuming higher speeds [37,38]. Figure 9, for example, shows for road Section 4 (km 31 + 950–km 32 + 450):

- the closure of direct accesses, introducing coordinated accesses and service roads;
- the widening of shoulders
- the removal of visual obstructions in curve 4 through vegetation cutting works;
- the introduction of centerline "rumble strips" at curve 4 and similar cross strips at approaches to some accesses and at intersections;
- the resurfacing of pavement and road markings, and the improvement of lighting conditions by installing an efficient lighting system at the approaches.

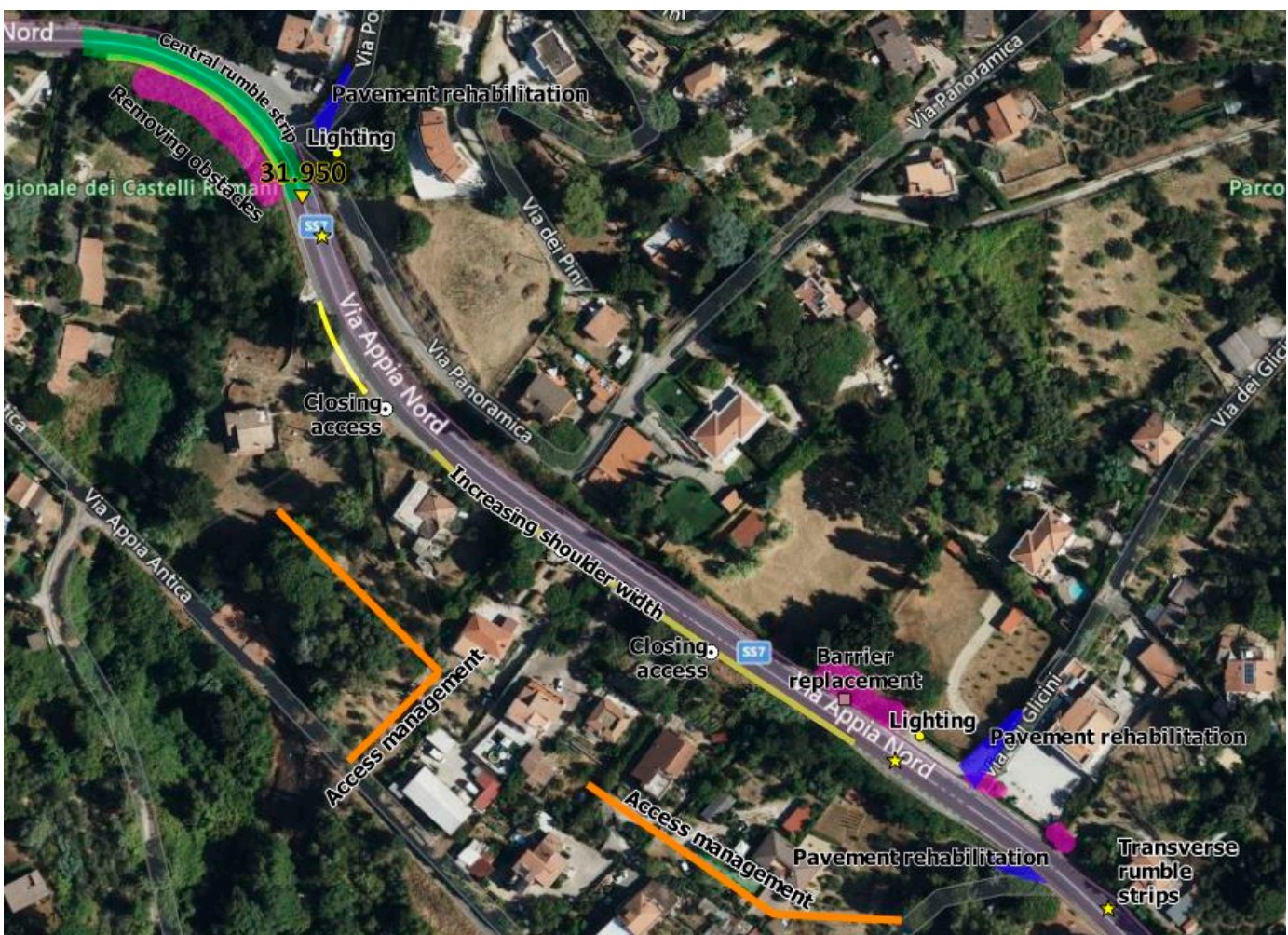

**Figure 9.** Location of the intervention measures provided for road Section 4.

## 5. Conclusions

From the inspection methods set out in Legislative Decree 35/2011 and explained in the Guidelines, some modifications and methodologies were proposed aimed at optimizing

inspection procedures and adapting them for secondary and local rural roads. Among the proposed criteria and methods, the combination of physical inspections with virtual inspections was suggested, through geo-referenced videos synchronized with a vehicle's location. These videos allowed for a global view of the infrastructure and an examination of all relevant details, carefully and repeatedly. The information that can be derived is therefore useful in the diagnosis phase, to recognize the risk factors induced by the presence of deficiencies and defects in the infrastructure. A particular important aspect for safety is to ensure the sight distance along the entire track, to allow the safe stopping maneuvers of vehicles at each point. By comparing this "required" distance with the unobstructed view offered by the road and the local conditions, any critical situations can be easily recognized and highlighted.

The inspection reports for freeways and highways were revised to consider the aspects that can actually be evaluated for secondary and local rural roads. In addition, a redefinition of the values and weights of the problems encountered was carried out according to criteria, as objectively as possible. Greater weights were assigned to risk factors mandatory with respect to safety requirements, and criteria were adopted based on the results of some studies in the literature, and carried out by observational, statistical and/or experimental methods.

Thus, the methodology proposed can promote greater objectivity in defining the outcomes of the inspections, limiting the weight of factors that may not actually constitute a risk to traffic, although characterized by a certain severity of judgment. In this way, the homogenization and uniformity of the inspection activities can be guaranteed, providing safer solutions for the inspectors, who are not required to exit their vehicles. In addition, the synthetic value assigned to the outcomes of the assessments allows for the results to be summarized, and returns a value to be compared with historical accident data. This comparison can be useful for hypothesizing and defining countermeasures against potential risk factors.

In this case study, the value $TS_{norm\_j}$, obtained from the report's elaboration, returned high scores in correspondence to higher accident rates. The typical accident patterns and dynamics are well related to the infrastructure safety issues encountered, and it was possible to deduce which interventions were best suited to mitigate the risks or effects of the highlighted deficiencies. Thus, the proposals for securing the infrastructure may be useful, effective, and cost-effective, achieving the maximum utility of safety inspections. Hereafter, this proposal could be repeatedly applied to a larger sample of different case studies, to adjust and adapt the index weights and compositions, depending on the territorial context.

**Author Contributions:** Conceptualization, C.C.B. and G.C.; methodology, C.C.B.; software, C.C.B.; validation, G.D.S., P.P. and G.C.; formal analysis, C.C.B.; investigation, G.C.; resources, G.D.S.; data curation, P.P.; writing—original draft preparation, P.P.; writing—review and editing, G.D.S.; visualization, C.C.B.; supervision, G.C.; project administration, G.C. All authors have read and agreed to the published version of the manuscript.

**Funding:** This research received no external funding.

**Data Availability Statement:** The data presented in this study are available on request from the corresponding author.

**Conflicts of Interest:** The authors declare no conflict of interest.

# Appendix A

| GENERAL INSPECTION Dir. Genzano | | | Dist. Initial (m) | 30+450 | 30+950 | 31+450 | 31+950 | 32+450 | 32+950 | 33+450 | 33+950 | 34+450 | 34+950 | 35+450 | 35+950 | 36+450 | 36+950 | 37+450 | Weights |
|---|---|---|---|---|---|---|---|---|---|---|---|---|---|---|---|---|---|---|---|
| | | | Dist. Final (m) | 30+950 | 31+450 | 31+950 | 32+450 | 32+950 | 33+450 | 33+950 | 34+450 | 34+950 | 35+450 | 35+950 | 36+450 | 36+950 | 37+450 | 37+950 | |
| Macro-item | Item | Parameter | Indicator | S1 | S2 | S3 | S4 | S5 | S6 | S7 | S8 | S9 | S10 | S11 | S12 | S13 | S14 | S15 | |
| General aspects | Traffic | Volume | suitability section | 2 | 2 | 2 | 2 | 2 | 2 | 2 | 2 | 2 | 2 | 2 | 2 | 2 | 2 | 2 | 1 |
| | | | special components | 2 | 1 | 2 | 1 | 1 | 2 | 1 | 2 | 2 | 2 | 3 | 2 | 1 | 2 | 2 | 1 |
| | Surrounding landscape | Roadside | disruption of marginal elements | 1 | 1 | 1 | 1 | 1 | 1 | 1 | 1 | 1 | 1 | 1 | 1 | 1 | 1 | 1 | 0.33 |
| | | | visual obstacles | 2 | 1 | 1 | 2 | 1 | 1 | 2 | 3 | 2 | 2 | 3 | 3 | 2 | 3 | 3 | 0.67 |
| | | | elements of distraction | 1 | 1 | 1 | 1 | 1 | 1 | 1 | 1 | 1 | 1 | 1 | 1 | 1 | 1 | 1 | 0.33 |
| | Speed | Speed data | project-allowed(+/-) | 1 | 1 | 1 | 2 | 2 | 1 | 1 | 3 | 3 | 3 | 1 | 3 | 3 | 2 | 0 | 1 |
| | | | operative-allowed (+/-) | 2 | 2 | 2 | 2 | 2 | 1 | 1 | 1 | 1 | 1 | 1 | 1 | 1 | 1 | 0 | 0.67 |
| | Geometry | Plan layout | geometry | 1 | 1 | 1 | 3 | 2 | 2 | 1 | 2 | 2 | 2 | 2 | 2 | 3 | 3 | 3 | 0.67 |
| Right-of-way | (Roadway, Median, and Roadside) | Shoulder | suitable width or absence | 3 | 3 | 2 | 1 | 1 | 2 | 2 | 2 | 1 | 2 | 3 | 3 | 3 | 2 | 2 | 1 |
| | | | shrinkage near a structure | N | N | N | N | N | N | N | N | N | N | N | N | N | N | N | 1 |
| | | Lane and Fast Lane | suitable width | 1 | 1 | 1 | 1 | 1 | 1 | 1 | 1 | 1 | 1 | 1 | 1 | 1 | 1 | 1 | 0.67 |
| | | | excess width | 1 | 1 | 1 | 1 | 1 | 1 | 1 | 1 | 1 | 1 | 1 | 1 | 1 | 1 | 1 | 0.33 |
| | | Roadway center visibility | divider/visibility to overtaking | 1 | 1 | 1 | 1 | 1 | 1 | 1 | 1 | 1 | 1 | 1 | 1 | 1 | 1 | 1 | 0.67 |
| | | Restraint devices | absence | 1 | 1 | 1 | 1 | 1 | 1 | 1 | 1 | 1 | 2 | 1 | 3 | 1 | 2 | 2 | 0.67 |
| | | | suitability | 1 | 0 | 1 | 1 | 0 | 1 | 0 | 0 | 0 | 1 | 0 | 0 | 1 | 1 | 1 | 0.67 |
| | | | transitions and terminals | 2 | 0 | 1 | 2 | 0 | 1 | 0 | 0 | 0 | 1 | 0 | 0 | 3 | 1 | 3 | 0.67 |
| | | | correct installation | 2 | 0 | 1 | 3 | 0 | 1 | 0 | 0 | 0 | 1 | 0 | 0 | 1 | 3 | 2 | 0.67 |
| | | | unprotected obstacles presence | 1 | 1 | 1 | 1 | 1 | 2 | 1 | 1 | 1 | 2 | 2 | 3 | 2 | 1 | 2 | 0.67 |
| | | Sideslope | building maintenance | 1 | 1 | 1 | 1 | 1 | 1 | 1 | 1 | 1 | 1 | 1 | 1 | 1 | 1 | 1 | 0.33 |
| | | | green maintenance | 1 | 2 | 2 | 2 | 2 | 3 | 2 | 2 | 2 | 1 | 2 | 1 | 3 | 2 | 1 | 0.33 |
| | | Drainage | maintenance | 3 | 1 | 3 | 1 | 1 | 1 | 3 | 3 | 1 | 1 | 3 | 1 | 1 | 1 | 1 | 1 |
| | | Fencing | maintenance | 1 | 3 | 3 | 3 | 2 | 1 | 1 | 1 | 2 | 1 | 1 | 1 | 1 | 1 | 3 | 0.33 |
| Road Markings | Pavement Marking | Visibility margin strips | day and night | 1 | 1 | 1 | 1 | 1 | 1 | 1 | 1 | 1 | 1 | 1 | 1 | 1 | 1 | 1 | 0.67 |
| | | visibility of lane marking strips | day and night | 1 | 1 | 1 | 1 | 1 | 1 | 1 | 1 | 1 | 1 | 1 | 1 | 1 | 1 | 1 | 0.67 |
| | | Special alerts | suitability or absence | 1 | 1 | 2 | 1 | 3 | 1 | 1 | 1 | 1 | 2 | 3 | 3 | 1 | 1 | 1 | 0.33 |
| | Vertical Signs | Vertical Signs | visibility | 1 | 1 | 2 | 1 | 1 | 2 | 1 | 1 | 1 | 2 | 2 | 1 | 3 | 3 | 3 | 0.33 |
| | | | readability | 1 | 1 | 1 | 1 | 1 | 1 | 1 | 1 | 1 | 1 | 1 | 1 | 3 | 3 | 3 | 0.33 |
| | | | intelligibility | 1 | 1 | 1 | 1 | 1 | 1 | 1 | 1 | 1 | 1 | 1 | 1 | 3 | 3 | 3 | 0.33 |
| | | Speed limit | correct positioning or absence | 1 | 1 | 1 | 1 | 1 | 1 | 1 | 1 | 1 | 1 | 1 | 1 | 1 | 1 | 3 | 1 |
| | | | speed detector | 1 | 1 | 1 | 1 | 1 | 1 | 1 | 1 | 1 | 2 | 1 | 1 | 1 | 2 | 1 | 1 |
| | | | operational speed adequacy | 1 | 1 | 1 | 1 | 1 | 1 | 1 | 1 | 1 | 1 | 1 | 1 | 1 | 1 | 1 | 1 |
| | Light Signals | danger, prescription and PMV | maintenance and absence | N | N | N | N | N | N | N | N | N | N | N | N | N | N | N | N |
| | | margin delineators | adequacy or absence | 1 | 1 | 1 | 1 | 1 | 1 | 1 | 1 | 1 | 1 | 1 | 1 | 1 | 1 | 1 | 1 |
| | Complementary Signs | curve delineators | adequacy or absence | 2 | 1 | 1 | 1 | 1 | 1 | 2 | 2 | 2 | 1 | 1 | 2 | 1 | 1 | 1 | 2 |
| | | margin delineators | adequacy or absence | 1 | 1 | 1 | 1 | 1 | 1 | 1 | 1 | 1 | 1 | 1 | 1 | 1 | 1 | 1 | 1 |
| Accesses | Accesses, Branches And Intersections | Coordination | adequacy | 1 | 2 | 3 | 3 | 1 | 2 | 3 | 1 | 2 | 1 | 3 | 3 | 2 | 1 | 3 | 1 |
| | | location of service and parking areas | adequacy | 1 | 1 | 1 | 1 | 1 | 2 | 1 | 1 | 1 | 1 | 1 | 1 | 1 | 1 | 1 | 1 |
| | | visibility | adequacy | 1 | 2 | 1 | 1 | 1 | 1 | 1 | 1 | 2 | 1 | 2 | 3 | 3 | 2 | 2 | 1 |
| Pavement | Surface | strains | presence | 1 | 1 | 1 | 1 | 1 | 1 | 2 | 1 | 1 | 1 | 1 | 2 | 1 | 2 | 1 | 1 |
| | | drainage | maintenance | 0 | 0 | 0 | 0 | 0 | 0 | 0 | 0 | 0 | 0 | 0 | 0 | 0 | 0 | 0 | 0 |
| | | adherence | suitability | 1 | 1 | 2 | 2 | 1 | 1 | 1 | 1 | 1 | 2 | 1 | 1 | 2 | 1 | 1 | 1 |
| | Joints | discontinuity | suitability | 1 | 1 | 1 | 1 | 1 | 1 | 1 | 1 | 1 | 1 | 1 | 1 | 1 | 1 | 1 | 1 |
| Other Aspect | Safety, Emergencies, and Interference | Parking And Emergency Pitches | adequacy and presence | 1 | 1 | 1 | 1 | 1 | 1 | 1 | 1 | 1 | 1 | 1 | 1 | 1 | 1 | 1 | 0.33 |
| | | Subservices | presence | 1 | 1 | 1 | 1 | 1 | 1 | 1 | 1 | 1 | 1 | 1 | 1 | 1 | 1 | 1 | 0.33 |
| | | Airlines | presence | 1 | 1 | 1 | 1 | 1 | 1 | 1 | 1 | 1 | 1 | 1 | 1 | 1 | 1 | 1 | 0.33 |
| | | | $TS_{GLj}$ | 16.02 | 11.33 | 18.31 | 18.69 | 8.99 | 12.33 | 13.32 | 16.67 | 15 | 19.68 | 22.34 | 24.69 | 22.69 | 21.02 | 25.37 | |

**Figure A1.** General Inspection Form dir. Genzano.

| Macro-item | Item | Parameter | Indicator | S1 (30+450 / 30+950) | S2 (30+950 / 31+450) | S3 (31+450 / 31+950) | S4 (31+950 / 32+450) | S5 (32+450 / 32+950) | S6 (32+950 / 33+450) | S7 (33+450 / 33+950) | S8 (33+950 / 34+450) | S9 (34+450 / 34+950) | S10 (34+950 / 35+450) | S11 (35+450 / 35+950) | S12 (35+950 / 36+450) | S13 (36+450 / 36+950) | S14 (36+950 / 37+450) | S15 (37+450 / 37+950) | Weights |
|---|---|---|---|---|---|---|---|---|---|---|---|---|---|---|---|---|---|---|---|
| General aspects | Traffic | Volume | suitability section | 2 | 2 | 2 | 2 | 2 | 2 | 2 | 2 | 2 | 2 | 2 | 2 | 2 | 2 | 2 | 1 |
| | | | special components | 2 | 1 | 2 | 1 | 1 | 2 | 1 | 2 | 2 | 2 | 3 | 2 | 1 | 2 | 2 | 1 |
| | Surrounding landscape | Roadside | disruption of marginal elements | 1 | 1 | 1 | 1 | 1 | 1 | 1 | 1 | 1 | 1 | 1 | 1 | 1 | 1 | 1 | 0.33 |
| | | | visual obstacles | 2 | 2 | 2 | 1 | 1 | 1 | 2 | 1 | 3 | 2 | 1 | 2 | 2 | 2 | 1 | 0.67 |
| | | | elements of distraction | 1 | 1 | 1 | 1 | 1 | 1 | 1 | 1 | 1 | 1 | 1 | 1 | 1 | 1 | 1 | 0.33 |
| | Speed | Speed data | project-allowed(+/-) | 1 | 1 | 1 | 2 | 3 | 1 | 1 | 3 | 3 | 3 | 1 | 3 | 3 | 2 | 3 | 1 |
| | | | operative-allowed (+/-) | 1 | 1 | 1 | 1 | 2 | 2 | 2 | 2 | 2 | 2 | 2 | 2 | 2 | 2 | 2 | 0.67 |
| | Geometry | Plan layout | geometry | 2 | 2 | 1 | 2 | 2 | 2 | 1 | 2 | 1 | 2 | 2 | 1 | 2 | 2 | 3 | 0.67 |
| Right-of-way | (Roadway, Median, and Roadside) | Shoulder | suitable width or absence | 2 | 2 | 2 | 3 | 2 | 3 | 2 | 3 | 2 | 1 | 3 | 3 | 3 | 3 | 3 | 1 |
| | | | shrinkage near a structure | N | N | N | N | N | N | N | N | N | N | N | N | N | N | N | 1 |
| | | Lane and Fast Lane | suitable width | 1 | 1 | 1 | 1 | 1 | 1 | 1 | 1 | 1 | 1 | 1 | 1 | 1 | 1 | 1 | 0.67 |
| | | | excess width | 1 | 1 | 1 | 1 | 1 | 1 | 1 | 1 | 1 | 1 | 1 | 1 | 1 | 1 | 1 | 0.33 |
| | | Roadway center visibility | divider/visibility to overtaking | 1 | 1 | 1 | 1 | 1 | 1 | 1 | 1 | 1 | 1 | 1 | 1 | 1 | 1 | 1 | 0.67 |
| | | Restraint devices | absence | 1 | 1 | 1 | 2 | 2 | 2 | 1 | 1 | 2 | 2 | 1 | 2 | 1 | 2 | 3 | 0.67 |
| | | | suitability | 1 | 1 | 1 | 1 | 1 | 1 | 1 | 1 | 1 | 1 | 1 | 1 | 1 | 1 | 1 | 0.67 |
| | | | transitions and terminals | 2 | 2 | 2 | 3 | 2 | 1 | 2 | 2 | 1 | 3 | 2 | 2 | 2 | 1 | 2 | 0.67 |
| | | | correct installation | 1 | 1 | 1 | 1 | 1 | 1 | 1 | 1 | 1 | 1 | 1 | 1 | 1 | 1 | 1 | 0.67 |
| | | | unprotected obstacles presence | 2 | 1 | 1 | 2 | 3 | 2 | 1 | 1 | 1 | 2 | 2 | 1 | 2 | 3 | 2 | 0.67 |
| | | Sideslope | building maintenance | 1 | 1 | 1 | 1 | 1 | 1 | 1 | 1 | 2 | 1 | 1 | 1 | 1 | 1 | 1 | 0.33 |
| | | | green maintenance | 1 | 2 | 2 | 3 | 1 | 2 | 2 | 1 | 2 | 2 | 1 | 2 | 1 | 2 | 1 | 0.33 |
| | | Drainage | maintenance | 3 | 1 | 3 | 3 | 1 | 1 | 3 | 3 | 1 | 1 | 3 | 1 | 1 | 1 | 1 | 1 |
| | | Fencing | maintenance | 1 | 3 | 3 | 3 | 2 | 1 | 1 | 1 | 2 | 1 | 1 | 1 | 1 | 1 | 3 | 0.33 |
| Road Markings | Pavement Marking | Visibility margin strips | day and night | 1 | 1 | 1 | 1 | 1 | 1 | 1 | 1 | 1 | 1 | 1 | 1 | 1 | 1 | 1 | 0.67 |
| | | visibility of lane marking strips | day and night | 1 | 1 | 1 | 1 | 1 | 1 | 1 | 1 | 1 | 1 | 1 | 1 | 1 | 1 | 1 | 0.67 |
| | | Special alerts | suitability or absence | 1 | 1 | 2 | 1 | 1 | 1 | 1 | 1 | 1 | 3 | 3 | 1 | 1 | 1 | 1 | 0.33 |
| | Vertical Signs | Vertical Signs | visibility | 1 | 1 | 1 | 1 | 1 | 1 | 1 | 1 | 3 | 1 | 1 | 1 | 3 | 1 | 1 | 0.33 |
| | | | readability | 1 | 1 | 1 | 1 | 1 | 1 | 1 | 1 | 3 | 1 | 1 | 1 | | 1 | 1 | 0.33 |
| | | | intelligibility | 1 | 1 | 1 | 1 | 1 | 1 | 1 | 1 | 3 | 2 | 1 | 1 | | 1 | 1 | 0.33 |
| | | Speed limit | correct positioning or absence | 1 | 1 | 1 | 1 | 1 | 1 | 1 | 1 | 1 | 1 | 1 | 1 | 1 | 1 | 1 | 1 |
| | | | speed detector | 1 | 1 | 1 | 1 | 1 | 1 | 1 | 1 | 1 | 1 | 2 | 1 | 1 | 1 | 1 | 1 |
| | | | operational speed adequacy | 1 | 1 | 1 | 1 | 1 | 1 | 1 | 1 | 1 | 1 | 1 | 1 | 1 | 1 | 1 | 1 |
| | Light Signals | danger, prescription and PMV | maintenance and absence | N | N | N | N | N | N | N | N | N | 3 | N | N | N | N | N | 0.33 |
| | | margin delineators | adequacy or absence | 1 | 1 | 1 | 1 | 1 | 1 | 1 | 1 | 1 | 1 | 1 | 1 | 1 | 1 | 1 | 0.33 |
| | Complementary Signs | curve delineators | adequacy or absence | 2 | 1 | 1 | 1 | 1 | 1 | 1 | 2 | 2 | 2 | 1 | 1 | 2 | 1 | 1 | 0.33 |
| | | margin delineators | adequacy or absence | 1 | 1 | 1 | 1 | 1 | 1 | 1 | 1 | 1 | 1 | 1 | 1 | 1 | 1 | 1 | 0.33 |
| Accesses | Accesses, Branches And Intersections | Coordination | adequacy | 1 | 2 | 3 | 3 | 1 | 3 | 3 | 1 | 1 | 1 | 3 | 3 | 2 | 1 | 3 | 1 |
| | | location of service and parking areas | adequacy | 1 | 1 | 1 | 1 | 1 | 1 | 1 | 1 | 1 | 1 | 1 | 1 | 1 | 1 | 1 | 0.33 |
| | | visibility | adequacy | 1 | 2 | 1 | 1 | 1 | 1 | 1 | 1 | 2 | 1 | 2 | 3 | 1 | 2 | 1 | 0.67 |
| Pavement | Surface | strains | presence | 1 | 1 | 1 | 1 | 1 | 1 | 2 | 1 | 1 | 1 | 1 | 2 | 3 | 2 | 1 | 0.33 |
| | | drainage | maintenance | 0 | 0 | 0 | 0 | 0 | 0 | 0 | 0 | 0 | 0 | 0 | 0 | 0 | 0 | 0 | 1 |
| | | adherence | suitability | 1 | 1 | 2 | 2 | 1 | 1 | 1 | 1 | 1 | 2 | 1 | 1 | 2 | 1 | 1 | 1 |
| | Joints | discontinuity | suitability | 1 | 1 | 1 | 1 | 1 | 1 | 1 | 1 | 1 | 1 | 1 | 1 | 1 | 1 | 1 | 0.33 |
| Other Aspect | Safety, Emergencies, and Interference | Parking And Emergency Pitches | adequacy and presence | 1 | 1 | 1 | 1 | 1 | 1 | 1 | 1 | 1 | 1 | 1 | 1 | 1 | 1 | 1 | 0.33 |
| | | Subservices | presence | 1 | 1 | 1 | 1 | 1 | 1 | 1 | 1 | 1 | 1 | 1 | 1 | 1 | 1 | 1 | 0.33 |
| | | Airlines | presence | 1 | 1 | 1 | 1 | 1 | 1 | 1 | 1 | 1 | 1 | 1 | 1 | 1 | 1 | 1 | 0.33 |
| | | | $TS_{GLj}$ | 15.02 | 13.01 | 18.99 | 23.01 | 15.03 | 16.02 | 16 | 17.68 | 20.64 | 19.03 | 23.69 | 23.34 | 20.68 | 19.03 | 22.03 | |

**Figure A2.** General Inspection Form dir. Velletri.

**PUNCTUAL INSPECTION**

Macro-item: **Flush Intersections/Coordinated Accesses**

Distance columns (Dist. Initial – Dist. Final, m) and associated roads (only columns containing data are shown below; all other 500 m segments between 30+450 and 37+950 are empty, scored 0):

| Item | Parameter | Indicator | 31+450–31+950 Via Appia Antica | 31+950–32+450 Via Poggio d'oro-Via panoramica | 32+450–32+950 Via dei Glicini - Via delle Fornaci | 33+950–34+450 Via della Faiola | 34+950–35+450 Via Colle dei Marmi - Via Acqua Lucia - Via Colle Noce | 36+450–36+950 Via Acqua Lucia | 36+950–37+450 Via Appia Vecchia | 37+450–37+950 Via Colle Caldara - Via Madonna degli Angeli - Via Appia Nord | 37+950 Via Ponte Veloce | Weights |
|---|---|---|---|---|---|---|---|---|---|---|---|---|
| Location And Conformation | Location | criticality | 1 | 3 | 1 | 1 | 1 | 3 | 1 | 2 | 3 | 0.67 |
| | | inadequate spacing from singular points | 1 | 2 | 2 | 1 | 1 | 3 | 2 | 2 | 3 | 0.67 |
| | Homogeneity Of Interfering Roads | appropriateness of the type of interfering road | 1 | 2 | 1 | 2 | 2 | 2 | 1 | 3 | 3 | 0.67 |
| | Intersection Spacing | appropriateness to context | 1 | 2 | 2 | 1 | 1 | 1 | 1 | 2 | 2 | 0.67 |
| | | appropriateness to the type of roadt | 1 | 2 | 2 | 1 | 1 | 1 | 1 | 2 | 2 | 0.67 |
| | Visibility | insufficient visibility from one or more entrance branches | 2 | 2 | 2 | 1 | 2 | 3 | 3 | 2 | 1 | 0.67 |
| | | insufficient visibility in particular lighting conditions | 1 | 3 | 3 | 3 | 2 | 3 | 3 | 3 | 2 | 0.67 |
| | | presence of elements that obstruct visibility | 1 | 2 | 2 | 1 | 2 | 3 | 1 | 1 | 1 | 0.67 |
| | Readability And Comprehension | difficulties in understanding by users | 2 | 2 | 2 | 1 | 1 | 1 | 2 | 2 | 1 | 0.67 |
| | | insufficiency of notice | 1 | 1 | 1 | | 1 | 1 | 1 | 3 | 3 | 0.67 |
| | | readability difficulties from one or more input branches | 2 | 1 | 1 | 1 | 2 | 1 | 1 | 2 | 1 | 0.67 |
| Geometry | Isles Of Channelization | disruption of marginal elements | N | N | N | N | N | N | N | N | N | 0.33 |
| | Accumulation Lanes | visual obstacles | N | N | N | N | N | N | N | N | N | 0.33 |
| | Analysis Of Allowed Maneuvers | adequacy and dangerousness | 2 | 2 | 2 | 1 | 2 | 3 | 3 | 3 | 2 | 0.67 |
| | Rotary | insufficient deflection | N | N | N | N | N | N | N | N | N | 0.33 |
| | | inadequate outer and/or inner diameter | N | N | N | N | N | N | N | N | N | 0.33 |
| | | unbalanced flows | N | N | N | N | N | N | N | N | N | 0.33 |
| | | insufficient capacity | N | N | N | N | N | N | N | N | N | 0.33 |
| Signaling System | Pavement Marking | adequacy of lane delineation | 3 | 2 | 3 | 3 | 2 | 3 | 3 | 3 | 3 | 0.67 |
| | | incorrect positioning arrows indication | 1 | 0 | 0 | 0 | 0 | 0 | 0 | 0 | 0 | 0.67 |
| | | adequacy of delimitation of non-transit zones | 0 | 0 | 0 | 0 | 0 | 0 | 0 | 0 | 0 | 0.67 |
| | | maintenance | 3 | 3 | 3 | 3 | 2 | 3 | 3 | 3 | 3 | 0.67 |
| | Vertical Signs | incorrect location | 1 | 2 | 2 | 3 | 1 | 1 | 1 | 2 | 1 | 0.67 |
| | | insufficiency or redundancy | 2 | 1 | 1 | 3 | 2 | 2 | 2 | 2 | 1 | 0.67 |
| | | maintenance | 1 | 1 | 1 | 3 | 1 | 1 | 1 | 1 | 2 | 0.67 |
| | Traffic Light | flow phasing adequacy | 0 | 0 | 0 | 0 | 0 | 0 | 0 | 0 | 0 | 0.33 |
| | | Adequacy phasing of conflicting vehicles | 0 | 0 | 0 | 0 | 0 | 0 | 0 | 0 | 0 | 0.33 |
| Other Aspect | Lighting | transition zones | 1 | 2 | 2 | 3 | 2 | 2 | 3 | 2 | 2 | 0.67 |
| | | insufficient lighting level | 1 | 2 | 2 | 3 | 2 | 2 | 2 | 2 | 2 | 0.67 |
| | | $TS_{Pt,i}$ | 10.72 | 22.11 | 19.43 | 17.42 | 14.74 | 21.44 | 18.09 | 26.8 | 21.44 | |
| | | $TS_{Pt,j}$ | 10.72 | 20.77 | | 17.42 | 14.74 | 19.77 | | 26.8 | 21.44 | |

**Figure A3.** Punctual Inspection.

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
