# Peer review of "Optimizing Road Safety Inspections on Rural Roads"

_infrastructures, doi:10.3390/infrastructures8020030_

Round 1

Reviewer 1 Report

The paper offers a definition of an objective approach to carry out safety inspections on rural roads on the basis of the procedures set out in Italian standards. Furthermore, the authors compared the results obtained for a case study road section with accident data analysis. The results show good consistency between the data obtained by implementing the two procedures.

The paper fits very well the topics of the journal, moreover, it should attract an audience in the field of road safety management. Therefore, I am happy to recommend publication.

My suggestions:

1)     I think that the indicator “green maintenance” (roadside) in Table 1 should not refer to the parameter “lane and fast lane”. Please, can you check it?

2)     Concerning the definition of an approach combining safety inspections and accident data analysis you can find interesting food for thought there:

§  Vaiana, R.; Perri, G.; Iuele, T.; Gallelli, V. A Comprehensive Approach Combining Regulatory Procedures and Accident Data Analysis for Road Safety Management Based on the European Directive 2019/1936/EC. Safety 2021, 7, 6. https://doi.org/10.3390/safety7010006

§  Vaiana, R.; Iuele, T.; Astarita, V.; Festa, D.C.; Tassitani, A.; Rogano, D.; Zaffino, C. Road safety performance assessment: A new road network Risk Index for info mobility. Procedia Soc. Behav. Sci. 2014, 111, 624–633.

3)     Please, upgrade the quality of figure 2.

4)     In Figure 3 information on the specific point is not clear. The image with the map could be greater, for example.

5)     I think that Figure 4 does not add information to the text in its current form. I suggest adding more useful information, such as the initial and final points of the inspection (e.g. the direction, Dir. Genzano, Dir. Velletri), e.g. the number of the element (this number should be reported in Table 2).

6)     … no more weaknesses!

Author Response

Review 1

The paper offers a definition of an objective approach to carry out safety inspections on rural roads on the basis of the procedures set out in Italian standards. Furthermore, the authors compared the results obtained for a case study road section with accident data analysis. The results show good consistency between the data obtained by implementing the two procedures.

The paper fits very well the topics of the journal, moreover, it should attract an audience in the field of road safety management. Therefore, I am happy to recommend publication.

We thank the reviewer for his/her careful reading of our manuscript and the many insightful comments.

My suggestions:

  • I think that the indicator “green maintenance” (roadside) in Table 1 should not refer to the parameter “lane and fast lane”. Please, can you check it?

We thank the reviewer for this statement. We realized that there was a mistake in the excerpt of the safety report, and we’ve modified it adding the “sideslope” parameter, as it can be seen in Table 1.

  • Concerning the definition of an approach combining safety inspections and accident data analysis you can find interesting food for thought there:
  • Vaiana, R.; Perri, G.; Iuele, T.; Gallelli, V. A Comprehensive Approach Combining Regulatory Procedures and Accident Data Analysis for Road Safety Management Based on the European Directive 2019/1936/EC. Safety 2021, 7, 6. https://doi.org/10.3390/safety7010006
  • Vaiana, R.; Iuele, T.; Astarita, V.; Festa, D.C.; Tassitani, A.; Rogano, D.; Zaffino, C. Road safety performance assessment: A new road network Risk Index for info mobility. Procedia Soc. Behav. Sci. 2014, 111, 624–633.

We appreciate the reviewer suggestions. We have found these 2 articles really interesting, we have carefully read them and inserted them in the text (See ref 13 and 26).

3) Please, upgrade the quality of figure 2.

We have replaced Figure 2 improving its quality.

4) In Figure 3 information on the specific point is not clear. The image with the map could be greater, for example.

We have replaced Figure 3 illustrating clearly information related to location and elevation referencing, speed, and grade. A yellow dot has been added in all the sub-images to highlight vehicle position along the road.

5) I think that Figure 4 does not add information to the text in its current form. I suggest adding more useful information, such as the initial and final points of the inspection (e.g. the direction, Dir. Genzano, Dir. Velletri), e.g. the number of the element (this number should be reported in Table 2).

We have replaced Figure 4 illustrating information related to road direction and basic geometric horizontal elements.

6) … no more weaknesses!

We thank the reviewer again for his/her support.

Reviewer 2 Report

2 Materials and methods - how much is optimized from traditional road inspections and how representative this single value quantifying the degree of safety can truly be? Discussion on costs, labor or time savings could be introduced or further discussed throughout the paper. 

Line 213:  if those indicators re removed from the reports, what are the possible implications in final analyses and results?

Line 301: Explanations on what the values for circular curves represent would be useful. In general, all the terms in row 2 of the table 2.

Line 371: widening of the shoulders would necessarily improve road safety performance?   It must be assessed in function of operating speeds, speed limits and desired speeds for the road under analysis.

In general, it is worth discussing further what are the limitations of this methodology in comparison to the traditional safety inspections. Are there contexts or any cases in which the video recordings do not provide the necessary information, or not with the necessary accuracy?  ¿What about night time recordings compared to day time ones?

The role of speed as one of the key factor risks in road safety analyses is also somehow missing. Infrastructure plays a key role in limiting or allowing circulation at certain speeds, and in general, a key component of speed management. 

Author Response

Review 2

2 Materials and methods - how much is optimized from traditional road inspections and how representative this single value quantifying the degree of safety can truly be? Discussion on costs, labor or time savings could be introduced or further discussed throughout the paper.

Firstly, we would like to thank the reviewer for his/her deep and careful reading of the paper.

In this paper, we propose a fast methodology that simplifies data acquisition, mining, and processing, which would require much more time and effort under traditional methods. The method involved the development of semi-automated calculation codes, which have a fixed structure that can be extended to other case studies. To date, we cannot quantify the representativeness of this unique index, as it is a proposal that has to be repeatedly applied to a larger sample of different case studies. In this way it will be possible to adjust and adapt the index weights and compositions, depending on the territorial context. In addition, the optimization consists mainly of an homogenization and uniformity of the inspection activities, providing also safer solutions for the inspectors, who are not required to get out of vehicles.

Line 213:  if those indicators are removed from the reports, what are the possible implications in final analyses and results?

We thank the reviewer for this alert. Actually, there was a bit of confusion about the simplification process. We have decided to modify the statement as follow, with the purpose of better explaining the procedure carried out and its implications: “Since it is not possible to fill the forms with such information, that kind of indicators have been removed from the reports, resulting in no change to the outcome.”

Line 301: Explanations on what the values for circular curves represent would be useful. In general, all the terms in row 2 of the table 2.

We thank the reviewer for this request. We have added the following statements to clarify the second row of Table 2 (see lines 315-328): “Where:

  • Lmin is the minimum length for a straight line, depending on the maximum design speed. In the specific case Lmin = 150 m;
  • Rmin is the minimum radius value for a specified road class. It depends on the minimum design speed of that class, and in the specific case Rmin= 118 m;
  • ΔV is the speed difference between two adjacent elements, characterized by their own design speed;
  • DT (transition distance) is the length in which the speed, according to the accepted theoretical model, passes from the value vi to vi+1, of two consecutive elements;
  • DR (recognition distance) is the maximum length of a road section within which the driver can recognize possible obstacles;
  • Rvmin is the minimum radius value for a vertical curve, related to the design criteria (i.e. geometric, dynamic, and sight distances verifications);
  • Δmax curve widening due to sight distances verifications.”

Line 371: widening of the shoulders would necessarily improve road safety performance?   It must be assessed in function of operating speeds, speed limits and desired speeds for the road under analysis.

To clarify the weight of the shoulder width, and also of the lane one, we investigated previous works and argued the discussion section with the following sentences and references (see lines 397-403):

“Several studies have recognized more cautious behavior of road users in the case of roads with narrow lanes and shoulders, as the driver is induced to reduce speed [35]. On the other hand, however, narrow shoulder may cause drivers to deviate their trajectory to the inside of the roadway, risking a head-on collision with oncoming vehicles, especially for rural roads where a median strip is absent [36]. Moreover, a shoulder width greater than 0.5 m allows users to perceive obstacles farther away and feel safer assuming higher speeds [37,38].”

In general, it is worth discussing further what are the limitations of this methodology in comparison to the traditional safety inspections. Are there contexts or any cases in which the video recordings do not provide the necessary information, or not with the necessary accuracy?  ¿What about night time recordings compared to day time ones?

The proposed methodology is an additional analysis, which does not claim to substitute traditional safety inspections, but it involves automation in the inspection processes, using most advanced instrumentation to simplify and streamline the conducted work. Moreover, in this way the outcome of the inspections is no more based on the experience and sensitivity of the inspector, due to his/her judgment uniformity.

We agree with the reviewer about the possible inaccuracy of the video recordings, which can provide biased information, depending on the sensitivity and quality of the recording device. However, in accordance with current regulation, we have performed several video recordings, both at night and daytime, to detect the infrastructure defects in different lighting conditions. In our method, video acquisition is a useful tool for objective and retrospective analysis, as it supports the operator and does not replace him in the inspection.

The role of speed as one of the key factor risks in road safety analyses is also somehow missing. Infrastructure plays a key role in limiting or allowing circulation at certain speeds, and in general, a key component of speed management.

In this study we follow the guidelines of the European directive, which focuses on the safety management of the infrastructure and not the entire road system. We agree with the reviewer about the importance and influence of speed in this kind of studies, but in this case we’ve decided to analyze just the infrastructure, to detect its defects to correct and restore properly, as required by the standards.

Reviewer 3 Report

1. Please define the variable m in Eq. (2)

2. Please clarify in Figures 6 and 7 the definition of the horizontal axis (x-axis)

Author Response

Review 3

We thank the reviewer for his/her careful reading of our manuscript.

  1. Please define the variable min Eq. (2)

We’ve reformatted the variable m at line 272, reporting it in italics in both equation and main text.

  1. Please clarify in Figures 6 and 7 the definition of the horizontal axis (x-axis)

We’ve replaced Figures 6 and 7, defining the name of the horizontal axis "Distance (m)" and improving the quality of the two images.
